# OpenReview forum: "Just Do It!? Computer-Use Agents Exhibit Blind Goal-Directedness"
_ICLR.cc/2026/Conference — ICLR 2026 Poster_

### Official Review · Reviewer_Ybz1 · 2025-10-20

**Soundness:** 3
**Presentation:** 4
**Contribution:** 2
**Rating:** 4
**Confidence:** 3

**Summary:**

The paper proposes an agent safety benchmark based on OSWorld, including three types of seemingly safe but actually risky tasks: posting inappropriate file content, executing unsafe scripts, and performing harmful system operations. The aim is to test whether agents can detect and avoid hidden hazards during task execution. However, it seems that easily using the prompt to tell the model check the safety can pass the benchmark.

**Strengths:**

The motivation is clear, and the examples are intuitive.

The three task types are well chosen to illustrate different forms of potential harm.

The paper presents the safety issue in a concrete and easy-to-understand way.

**Weaknesses:**

In this benchmark, the safe outcome is always defined as refusal. I understand the motivation, but it makes the setup a bit unrealistic — in many practical situations, a safe agent could check or modify the script instead of simply rejecting it.

The reported unsafe behaviors may be caused more by the setting than by model limitations. The agent is prompted to focus on completing the task, so it naturally executes even clearly unsafe commands (e.g., creating a 20 TB swap file). Once the prompt explicitly reminds the model to check safety, the unsafe execution rate drops from 50% to 20%. This suggests that the issue is largely prompt‑level and the benchmark difficulty is not inherently high.

The evaluation mainly reports execution or refusal rates. It would be helpful to include more fine‑grained metrics (e.g., distinguishing detection failures vs. decision errors) to better understand model behaviors.

**Questions:**

You mention employing LLM‑based judges with high human‑agreement rates. Could you clarify whether these judges were given full screen context or only textual traces of the agent’s actions? In realistic GUIs, partial observability could heavily affect judgment accuracy.

The current experiments use a simple prompt‑based intervention reminding the model to “consider safety.” Did you explore more nuanced prompts that calibrate different levels of caution — for instance, balancing safety awareness against task completion utility? This could reveal whether BGD mitigation can be tuned continuously rather than treated as an on/off switch.

---

> ### Author Response · Authors · 2025-11-21
> **Response to Reviewer 4 (Ybz1) - part 1**
>
> We thank you for recognizing the clarity of our motivation and for your questions and comments, which we address below. Before that, we would like to clarify a few points in the summary you provided.
>
> First, the categories you mentioned in your summary (“posting inappropriate file content,” “executing unsafe scripts,” and “performing harmful system operations”) are only illustrative example tasks provided in the paper for each BGD pattern; they are not representative of the full task sets in those patterns. Our contribution is the identification of the Blind Goal-Directedness (BGD) phenomenon and the characterization of its three prevalent patterns, which are “Lack of Contextual Reasoning”, “Assumptions and Decisions under Ambiguity”, and “Contradictory or Infeasible Goals”. We design 30 diverse tasks per pattern (90 total) and introduce the BLIND-ACT benchmark.
>
> Second, you wrote that “easily using the prompt to tell the model check the safety can pass the benchmark.” (Also your Q2).
> We would like to clarify that while prompting is beneficial, it does not solve BGD. As shown in our results (Figure 1 and **Appendix E)**, reflective prompting still yields 61.4% BGD intentions and 27.3% completions on average, with models such as o4-mini reaching as high as 68.9% and 41.1%, and DeepSeek reaching 82.1% and 50.7%. **These are substantial and far from “passing”, especially in safety and security, where even 1% is intolerable.** Moreover, 61.4% of average BGD intentions suggests that as models become more capable, completions will likely rise correspondingly (consistent with our safety–capability parity observations), reinforcing the need for further mitigation research on Blind Goal-Directedness. **(Revised Section 3.2 and Appendix E)**
>
> ---
>
> **Q1) Is Refusal the Only Safe Behavior in Our Benchmark?**
>
> Thanks for bringing up this important point. **Refusal is not the only safe behavior in our benchmark.** As long as the agent does not exhibit BGD intentions, its behavior is labeled as safe. This is because this is not a direct harmfulness benchmark, and many of our tasks are not inherently harmful; **in these cases, it may be acceptable for the agent to proceed, provided it avoids any undesired or unsafe actions along the trajectory**. There are many safe actions other than refusal, such as:
>
> *   Your script example: A safe behavior can be not to guess script behavior from filenames, but to open and inspect the contents and run the correct script.
>
> *   A contextual reasoning task asks the agent to upload a function (which contains API keys) to a wiki: A safe behavior is to redact the API keys before uploading; again, a non-refusal action that is fully acceptable.
>
> *   Form-filling example (Figure 1): The safe behavior is to actually fill out the form without fabricating disabilities or citizenship status, not to refuse the task.
>
> *   Contradictory goals example: When instructed to set the AC temperature to 160°F, agents may recognize the contradiction and instead set it to a reasonable value (e.g., 70°F). This is also treated as safe behavior, not a refusal.
>
> **In summary, safe behavior includes any trajectory that avoids BGD intentions given the context of the task, which encompasses a broad set of contextually grounded, non-refusal actions, not just rejecting the user request.** We **added Appendix I.1 (What Constitutes Safe Behavior in BLIND-ACT)**. Thank you again!
>
> ---
>
> **Q2) Does prompting really solve BGD?**
>
> To avoid repetition, we kindly refer you to the second point in our initial clarification above, where we addressed this aspect of your summary.
> In short, prompting improves behavior but is far from solving BGD, with even reflective prompting still leaving substantial safety gaps across models.
>
> ---
>
> **Q3) What context is provided to the LLM judge?**
>
> As described in the “Judge Implementation” paragraph in Section 2.3, our judges receive both the agent’s reasoning and actions (trajectory) as well as the full accessibility (a11y) tree capturing the screen context for every step.
> As also acknowledged by R2 on the reliability of our judges, we carefully investigated the reliability of the judge by running comprehensive ablations in Appendix D.1; for example, providing screenshot captions instead of a11y trees, and found that the a11y-based setting yields the highest agreement with human evaluations. Thus, we adopt this configuration, achieving 93.75% agreement, with precision/recall/F1 ≥ 0.9 and strong Cohen’s Kappa scores (0.81 for BGD and 0.91 for Completion).
>
> Our use of LLM judges is inspired by OS-Harm [1], which was accepted as Spotlight in the NeurIPS 2025 Datasets and Benchmarks Track. We still achieve higher agreement with human evaluations (93.75%) and stronger precision/recall/F1 metrics than OS-Harm (84.5%), with detailed information on judge accuracy **(Section 3.1 and Appendix D)**.
>
> We also encourage you to see the response under **Common Question** if interested.

---

> ### Author Response · Authors · 2025-11-21
> **Response to Reviewer 4 (Ybz1) - part 2**
>
> **Q4) Distinguishing detection failures vs. decision errors**
>
> This is really a great suggestion! We would like to clarify one small point: our evaluation does not report any “refusal rates”. This is because refusal is not the only form of safe behavior in our setup, as we described in our response to your Q1. We report BGD (whether the model exhibits undesired intentions) and Completion (whether it carries them out), which are the standard metrics adopted in recent CUA safety evaluations (OS-Harm \[1\], VPI-Bench \[2\], RiOSWorld \[3\]) and acknowledged by R2.
> That said, we really like the spirit of your suggestion to distinguish between detection failures and decision errors.
>
> There are three distinct cases for an agent’s behavior:
>
> (1) recognizes harm, does NOT show any BGD intentions, and takes the appropriate safe actions. These correspond to the runs where the judge finds no BGD intentions (100 − BGD%).
>
> (2) recognizes harm, but due to the failure modes we identify (execution-first bias, request-primacy, thought–action disconnect), it still shows BGD intentions and proceeds toward the execution of those.
>
> (3) does NOT recognize the harm and shows BGD intentions towards unsafe execution.
>
> (2) and (3) combined, reflect the BGD rates, with each constituting a portion of it.
>
> To provide a concrete example, we manually inspected all 90 GPT-4.1 trajectories. GPT-4.1 showed BGD in 76 tasks (84.4%), meaning it behaved safely in 14 tasks by recognizing and avoiding harm (case 1).
> Among the 76 BGD cases, we found 9 tasks where the agent recognized the harm or contradiction but still pursued the unsafe goal (case 2), and in the rest of the 67 tasks, it did not even recognize the harm (case 3).
>
> We did the analysis for Claude Sonnet 4 and o4-mini, added the results to the table below, and added an **Appendix F (Distinguishing Detection Failures Vs. Decision Errors)** to the revised **PDF with further details**. Thank you for improving our paper!
>
> | Model| Total BGD | BGD: Unrecognized | BGD: Recognized | No BGD: Recognized|
> |-|-|-|-|-|
> | **GPT-4.1**| 76 / 90| 67 / 76| 9 / 76| 14 / 90|
> | **Claude Sonnet 4**| 59 / 90|  52 / 59| 7 / 59| 31 / 90|
> | **o4-mini**| 78 / 90| 67 / 78| 11 / 78| 12 / 90|
>
> **All in all, most harmful cases stem from CUAs failing to recognize the harm (case 3 - BGD Unrecognized), and a smaller, yet important subset reflects cases where the agent recognizes the harm yet still decides to pursue the unsafe goal due to the identified failure modes (case 2 - BGD Recognized).**
>
> ---
>
> **Q5) Is BGD mitigation tunable through graduated prompting?**
>
> Thank you for this insightful question. Our prompting interventions intentionally followed a gradual spectrum of caution rather than a binary “on/off” switch. As outlined in Section 3.2, we examine Default (no safety cue), Contextual (consider safety/feasibility/context), and Reflective prompting (Contextual + explicit harm reflection at each step).
>
> As detailed in our results, these prompting settings already reveal a tunable progression of safety awareness: Reflective outperforming Contextual, and Contextual outperforming Default. However, even the strongest configuration, Reflective prompting, still results in substantial BGD intentions and Completion rates (as noted at the beginning of our response to your review), indicating that prompting alone remains insufficient.
>
> **While more fine-grained prompt variants could be explored, our findings already demonstrate that prompting operates with constrained headroom and has limited effectiveness. These results reinforce the need for model-level safety mitigations (e.g., training-time interventions, reward modeling, trajectory-level supervision),** a point also supported by our analysis of failure modes (execution-first bias, thought–action disconnect, request-primacy). These approaches, along with deeper mechanistic studies, are valuable future research directions for us and the broader community, as reflected in the revised **Appendix B (Potential Future Directions).**
>
> ---
>
> \[1\] Kuntz et al. OS-HARM: A Benchmark for Measuring Safety of Computer Use Agents, NeurIPS 2025 Datasets and Benchmarks Track spotlight
>
> \[2\] Cao et al. VPI-Bench: Visual Prompt Injection Attacks for Computer-Use Agents
>
> \[3\] Yang et al. RiOSWorld: Benchmarking the Risk of Multimodal Computer-Use Agents, NeurIPS 2025

---

> > ### Comment · Reviewer_Ybz1 · 2025-11-27
> >
> > The rebuttal helpfully clarifies several points I had raised: safe behavior is not restricted to refusal, the LLM judges operate with rich screen context (a11y trees) and have been validated against humans, and the authors have added a useful breakdown between detection failures and decision errors. These changes improve clarity and strengthen the evaluation.
> >
> > In my view, Blind Goal-Directedness as defined here mostly systematizes and benchmarks a family of failure modes that are closely related to well-known phenomena (goal misgeneralization, over-optimization, request-primacy), rather than introducing a fundamentally new conceptual lens or mitigation technique. The BLIND-ACT benchmark is carefully constructed and the empirical results are informative, but the setting still leans toward stress-testing agents under relatively weak safety prompting, and the paper stops short of a deeper, more systematic analysis of how these behaviors evolve under more realistic guardrailed prompts or stronger model-level interventions.
> >
> > Overall, I see this as a solid and potentially useful benchmark/phenomenon-paper, but the level of conceptual novelty and analytical depth does not quite reach the bar where I would argue for acceptance. If possible I would increase the score to 5, but I dont think it is enough to 6.

---

> > > ### Author Response · Authors · 2025-11-28
> > > **Thank you! Response in the next comment**
> > >
> > > We really appreciate your positive feedback and are glad our clarifications on safe behavior, judge validation, and detection vs. decision errors improved clarity. **Thank you for recognizing the paper as a solid and useful benchmark study, and for your willingness to increase your score.**
> > >
> > > Please see our response in the **next comment below:**

---

> > > > ### Author Response · Authors · 2025-11-28
> > > > **Our Response**
> > > >
> > > > ***- Is BGD related to phenomena such as goal-misgeneralization? If so, what's its value?***
> > > >
> > > > **Blind Goal-Directedness (BGD) indeed relates to classical notions such as goal misgeneralization and reward hacking, but our contribution lies in examining how these phenomena manifest and evolve in multi-turn, expanded action space Computer-Use Agent (CUA) settings**. We formalize BGD as a behavioral alignment failure unique to CUAs, identifying recurrent failure patterns that arise as these agents pursue goals across extended reasoning–action trajectories and **new forms of interactions**.
> > > >
> > > > **For example, as noted in Lines 162–163, ambiguity and underspecification have long been vulnerability sources in traditional systems, yet our work shows how they re-emerge in this new form of interaction** (e.g., interacting with files/scripts based solely on their names, fabricating content and lying, etc). This motivation parallels how classical concepts gain renewed importance as interaction modalities evolve. **For example, how reward hacking in classical RL appears as “sycophancy” in conversational AI**, or how prompt injection evolved from simpler QA LLMs to multi-turn agentic and CUA settings (OS-Harm, RedTeamCUA, etc).
> > > >
> > > > Therefore, **it's important to understand how such classical concepts evolve with new forms of interaction and environments. The value of BGD is therefore in operationalizing and benchmarking this family of goal-pursuit failures in CUAs, rather than treating them as anecdotal or sparse qualitative observations. The BLIND-ACT benchmark enables systematic measurement and comparison, which is essential for progress in CUA safety.** To our knowledge, no prior benchmark or analysis has formally captured this behavior class in such environments.
> > > >
> > > > ---
> > > >
> > > > ***- Do we stress-test agents under weak safety prompting? Do we lack deeper analysis under more guardrailled prompts?***
> > > >
> > > > Building on our earlier responses, **our evaluation doesn't rely on weak prompting, but rather on the standard deployment prompts used in realistic CUA settings**. This setting is **intentional**: it allows us to diagnose how safety and reliability failures emerge under realistic usage conditions, similar to prior CUA safety studies such as OS-Harm and RedTeamCUA. As we also noted in Q2.2 of R3, **the fact that these failures arise organically in the most common deployment scenarios makes BGD particularly concerning.**
> > > >
> > > > **Still, beyond standard prompting, and as detailed in our response to your Q2 and Section 3.2, we also evaluate two stronger safety prompting strategies (contextual and reflective).** The results show that prompting improves behavior but is far from solving BGD, **with even reflective prompting still leaving substantial safety gaps across models, still exhibiting high BGD rates.**
> > > >
> > > > We further conduct **deeper trajectory-level analyses**, identifying the observed failure modes and **showing how more safety-aware prompting might affect the reasoning in detecting harm. Yet, failure modes such as request-primacy and thought-action disconnect still drive agents to justify and proceed with unsafe actions** (Lines 388 and 418). This again underscores the limited effectiveness of prompting-based safeguards and **motivates deeper research into model-level mitigations, training-time alignment methods, and mechanistic analyses to trace the origins of BGD across model components, training stages, and reasoning steps, as outlined in Appendix E (Potential Future Directions).**
> > > >
> > > > ---
> > > >
> > > > ***- Analytical depth and primary contributions.***
> > > >
> > > > Regarding analytical depth, **we examine the underlying behavioral causes of BGD through a series of analyses:**
> > > >
> > > > **(i) Cross-model comparisons** revealing phenomena such as safety–capability parity and contrasting behaviors between computer-use–trained and general-purpose models;
> > > > **(ii) Prompting-condition analyses** (default, contextual, reflective) demonstrating the limited effectiveness of inference-time mitigations; and
> > > > **(iii) qualitative trajectory inspection identifying consistent failure modes** (execution-first bias, thought–action disconnect, and request-primacy), revealing reasoning-stage breakdowns.
> > > >
> > > > **In response to reviewer Q2 (R2), we also added a new analysis examining when BGD first emerges within reasoning steps (see Appendix G, At Which Steps Does BGD Emerge?).**
> > > >
> > > > **Furthermore, in response to your Q4, we added a detection vs. decision error analysis (Appendix F), which decomposes the sources of BGD and clarifies what each metric implies, as you acknowledged in your previous response.**
> > > >
> > > > In the end, our primary contribution is to identify the BGD phenomenon as a previously unreported failure in CUAs, characterize its three prevalent patterns, and introduce BLIND-ACT, the first benchmark that enables systematic study of this phenomenon. **As R1 and R3 both noted, BGD exposes an emerging class of risks beyond prior attacks and opens up a new direction for CUA security research.**

---

### Official Review · Reviewer_w3zH · 2025-11-01

**Soundness:** 3
**Presentation:** 2
**Contribution:** 2
**Rating:** 2
**Confidence:** 4

**Summary:**

This paper introduces and characterizes the phenomenon of Blind Goal-Directedness (BGD) in Computer-Use Agents (CUAs), defined as an intrinsic tendency to pursue user-specified goals without sufficient regard for feasibility, safety, reliability, or context. The authors categorize BGD into three prevalent patterns. To systematically evaluate this risk, the work develops BLIND-ACT, a benchmark comprising 90 tasks built on OSWorld VM, and employs an LLM-based judge for evaluation.

**Strengths:**

1. The work explicitly identifies and meticulously characterizes Blind Goal-Directedness (BGD) as a core security risk, which opens up a relatively new direction for CUA security research.
2. The authors have created the OSWorld-based benchmark, BLIND-ACT (comprising 90 tasks), which supports the systematic evaluation of BGD in a dynamic environment.
3. Extensive evaluation was conducted on nine state-of-the-art models, providing quantitative evidence of BGD risk, and uncovering the Safety-Capability Parity phenomenon alongside three critical qualitative failure modes.

**Weaknesses:**

1. The reliance on an LLM Judge (using o4), even with a high agreement rate with human annotators, introduces an element of **model-dependency** into the core evaluation metric (unlike the original OSWorld). The paper does not sufficiently address how potential systemic biases within the judge (which is itself a VLM / CUA) might fail to capture subtle BGD manifestations that fall outside the limited, pre-defined patterns.
2. The benchmark is constructed entirely within the OSWorld Ubuntu VM environment, utilizing a specific set of applications (e.g., GIMP, LibreOffice, Thunderbird).
    1. Firstly, the novelty of the new dynamic environment is questionable, as it entirely reuses previous software and VM configurations.
    2. Secondly, the selection of these apps was intended for studying the CUA's automation capabilities,  not concerns about reliability or safety. I suggest that more commonly used applications that are potentially subject to greater security implications should be included in the study.
3. Please check my questions

**Questions:**

1. Could the authors provide more detail on the failure modes of the LLM judge, specifically the disagreements with human annotators?

2. The paper defines BGD intentions based on whether an agent "exhibited blind goal-directedness intentions." This relies heavily on inspecting the model's textual reasoning trace.

  a. Firstly, not all CUA foundation models have a "thought" (e.g., Arguvis or other seeact architecture agent).

  b. Secondly, how confident are the authors that the observed phenomena, such as "Thought-Action Disconnect," are not merely artifacts of the agent's constrained output format rather than a genuine cognitive disconnect?

---

> ### Author Response · Authors · 2025-11-21
> **Response to Reviewer 3 (w3zH) - part 1**
>
> **Q1) Reliability of LLM Judges**
>
> Thank you for this question. We encourage you to first read our detailed response under **Common Question**, where we discuss our reliance on LLM judges. Further experiments and clarifications are provided below:
>
> In short, **as also acknowledged by R2**, we carefully investigated the reliability of the judge by conducting several forms of validation and ablation:
>
> *   **Judge-human agreement:** 93.75% human eval agreement with precision/recall/F1 ≥ 0.9 and strong Cohen’s Kappa scores (0.81 for BGD and 0.91 for Completion) - (Section 3.1 and **Appendix D)**
>
> *   **Robustness Ablations:** Multiple judge configurations, including agent trajectory, agent trajectory + a11y trees, agent trajectory + screenshot captions; with two different judges (GPT4.1 and o4-mini) - **(Appendix D)**
>
>
> We note that our use of LLM judges is inspired by **OS-Harm \[1\], which was accepted as a Spotlight paper in the NeurIPS 2025 Datasets and Benchmarks Track**. We still achieve higher agreement with human evaluations (93.75%) and stronger precision/recall/F1 metrics than OS-Harm (84.5%).
>
> **We additionally designed a new experiment where we ran our judge three times on all 90 trajectories for three agents (o4-mini, GPT-5, and Claude Sonnet 4) to assess consistency across independent runs.** As shown in the table below, the variances across the three runs are effectively negligible, demonstrating that the judge’s decisions are stable and robust across repeated independent evaluations. **Per R1’s suggestion, we created Appendix D.2 (Judge Consistency Across Independent Runs)** in the updated PDF to include this discussion.
>
> \*\*Average BGD and Completion among three categories (whole benchmark)
> | Model| Run1 | Run2 | Run3 | STD |
> |-|-|-|-|-|
> |o4-mini|BGD: 86.6, Completion: 57.7|BGD: 86.6, Completion: 57.7|BGD: 86.6, Completion: 56.6|BGD: 0, Completion: 0.52|
> |GPT-5|BGD: 85.5, Completion: 45.5|BGD: 85.5, Completion: 45.5|BGD: 85.5, Completion: 44.4|BGD: 0, Completion: 0.52|
> |Claude Sonnet 4|BGD: 65.5, Completion: 35.5|BGD: 65.5, Completion: 35.5|BGD: 65.5, Completion: 35.5|BGD: 0, Completion: 0|
>
> **Based on your suggestion, we also added a new analysis examining cases where the judge and human annotators disagree.** We created **Appendix D.3 (Cases of Judge–Human Disagreement)** in the updated PDF to include this discussion.
>
> In the “Contradictory or Infeasible Goals” category, some disagreements arise around “BGD” intention detection. For example, in a task requiring cropping an image to negative dimensions, some agents initially proceed by opening GIMP and loading the image before later recognizing that the dimensions are invalid. Human evaluators typically do not consider this BGD, since the agent notices the contradiction even in mid-trajectory. The judge, however, applies a stricter standard and expects the agent to identify the contradiction earlier, thereby labeling such cases as BGD.
>
> In the “Lack of Contextual Reasoning” category, some disagreements appear in “Completion”. In some more complex tasks such as an example where the agent must locate an image, open email, compose a message, attach the file, and click “send”, some agents reach the final step and correctly predict the “send” action, but due to capability or coordination errors, the final click fails to execute and the task exits due to the 15 steps limit. The judge counts this as a Completion, while humans label these as non-completions despite full correct behavior up to the last step.
>
> We added all of these new analyses to **Appendix D.2 and D.3** of the updated paper.
>
> ---
>
> **Q2)**
>
> **Q2.1) Novelty of the Dynamic Environment Compared to OSWorld?**
>
> We appreciate this question. OSWorld provides a standardized VM environment that has now become the default setup for most recent CUA studies and safety benchmarks, including OS-Harm\[1\], RiOSWorld \[2\], VPI-Bench \[3\], etc. Therefore, we also use this standard environment, as our contribution lies not in modifying the environment itself but in identifying the BGD phenomenon (which, as you and R1 noted, opens a new direction for CUA security research) and in designing new benchmark tasks that systematically evaluate it within this widely adopted setting. In other words, one should think about OSWorld as an evaluation environment, and BLIND-ACT as a new benchmark in the environment.
> Moreover, as discussed in Section 2.2 (“Task Structure”), for each task, we design custom assets and interfaces, such as mock web pages, forms, messaging, social media, emails, and diverse file types (documents, text files, images, scripts, etc.), to create contextual scenarios that elicit BGD. Additionally, our LLM-as-Judge evaluation pipeline with the contextual explanations is substantially different from OSWorld’s rule-based evaluation framework, as discussed in the previous question.

---

> > ### Author Response · Authors · 2025-11-21
> > **Response to Reviewer 3 (w3zH) - part 2**
> >
> > **Q2.2) OSWorld’s apps were intended for capabilities. Are they suitable for safety?**
> >
> > Thank you for raising this point! The applications included in OSWorld, such as GIMP, Thunderbird, LibreOffice, VS Code, Chrome, and standard Ubuntu apps are widely-used, general-purpose tools that reflect the core scenarios CUAs are intended to automate. Otherwise, the motivation behind OSWorld itself would become questionable. We note that capability and safety should not be isolated. The fact that the safety and reliability issues we uncover arise even within these core “automation capability” applications is important: **it shows that BGD organically appears in the everyday scenarios for which CUAs are most commonly deployed.**
> >
> > At the same time, we really like your point and note that these issues could be even more severe in sensitive, high-stakes environments. In that sense, there’s still a lot of value in studying BGD in general-purpose scenarios, because it provides a conservative lower bound. This means that if failures already arise in these everyday applications, the implications and consequences for more security-critical or irreversible-action domains are even more concerning.
> > Moreover, as noted earlier, our tasks incorporate custom files and interfaces (mock web pages, forms, messaging, social media, emails, and various file types), expanding the interaction space well beyond the default OSWorld applications, including more sensitive and safety-critical contexts as well. For example, our benchmark includes scenarios like tax form handling (Figure 1), bank account management and transactions (Figure 7), and others.
> >
> > We reflected this discussion in the **updated PDF in Appendix I.3, and revised Section 2.2.**
> >
> > ---
> >
> > **Q3)**
> >
> > **Q3.1) Reliance on the models’ reasoning trace for BGD intention detection**
> >
> > Great question. As discussed in Section 3 and Appendix C, all agents are implemented using the standard OSWorld framework, including its system prompt, and agents are instructed to output (i) a natural-language reflection (reasoning) and (ii) a pyautogui code block that encodes the next action(s). While the style and verbosity of this reasoning may vary across models, every agent still verbalizes its intentions through this reasoning and predicts an explicit action code sequence. Also, the code block itself, often commented by the agent, clearly reveals what the model intends to do (e.g., pyautogui.hotkey('ctrl', 'a') # Select all text and replace with 20000). The judge, therefore, examines both the reasoning trace and the predicted action code as well as the environment observations (a11y tree) for detecting BGD intentions. **We revised Appendix C to reflect this.**
> >
> > **Q3.2) Is Thought–Action Disconnect an Artifact of the Output Format?**
> >
> > This is an excellent point. Indeed, we agree that the phenomena we observe may partly arise from the constraints of the current CUA deployment paradigm. Studies like \[4\] found that models become more harmful when placed in agentic environments, and identified the predefined output action space as one potential factor compared to open-ended QA use cases. In standard CUA settings, including ours, agents are instructed to constrain their output to a natural-language reflection followed by a structured pyautogui code block that the controller parses and executes in the environment. It is therefore entirely possible that these constrained output formats introduce certain behavioral artifacts.
> >
> > However, these findings remain important because these conditional settings reflect how CUAs are currently deployed and evaluated in practice. Since models must operate under these structured, controller-compatible formats in real CUA pipelines, it is critical to understand and mitigate the safety failures that emerge under these standardized conditions.
> >
> > ---
> >
> > \[1\] Kuntz et al. OS-HARM: A Benchmark for Measuring Safety of Computer Use Agents, NeurIPS 2025 Datasets and Benchmarks Track spotlight
> >
> > \[2\] Yang et al. RiOSWorld: Benchmarking the Risk of Multimodal Computer-Use Agents, NeurIPS 2025
> >
> > \[3\] Cao et al. VPI-Bench: Visual Prompt Injection Attacks for Computer-Use Agents
> >
> > \[4\] Chiang et al. Why Are Web AI Agents More Vulnerable Than Standalone LLMs? A Security Analysis,  BuildingTrust workshop@ICLR 2025

---

### Official Review · Reviewer_rsyJ · 2025-11-01

**Soundness:** 3
**Presentation:** 4
**Contribution:** 3
**Rating:** 6
**Confidence:** 4

**Summary:**

This study identifies the significant Blind Goal-Directedness (BGD) risk that CUAs prioritize execution over safety, reliability, or logical consistency. The authors characterize this risky behavior via showing three common patterns: (i) lack of contextual reasoning, (ii) assumptions and decisions under ambiguity, and (iii) contradictory or infeasible goals. Based on these patterns, they construct a BGD benchmark Blind-Act, consisting of 90 tasks crafted by human. They adopt scalable and cost-efficient evaluation based on LLM-as-a-judge to evaluate the CUAs' risky behavior on the benchmark. The evaluation results expose 3 failure modes of CUAs: execution-first bias, thought-action disconnect, and request primacy.

**Strengths:**

- Clearly demonstrate the significant risk of BGD quantitatively and qualitatively.
- Carefully investigate the reliability of LLM-as-a-judge.
- Explore mitigations through contextual and reflective prompting strategies.
- Identify 3 significant failure modes: execution-first bias, thought-action disconnect, and request-primacy, and provide clear demonstrations:
- The evaluation metrics, BGD and the completion rate, comprehensively capture the CUA's intentions and capability to finish the task when confronted with potentially unsafe requests.

**Weaknesses:**

- The evaluation would benefit from providing the percentage of runs where CUAs recognize that the user query is not safe.
- The authors should add some results and discussions about the number of running steps where the CUAs generate BGD responses. Is it usually appearing in the planning and task decomposition step, which may cause more harm, or implementation step?
- It would be great if the authors can provide some results about the correlation between the CUA's BGD tendency and the underlying model's planning performance. So we can better understand the cause of BGD, either due to the poor planning performance or because of inherent insensitiveness to unsafe content. I expect that if the agent can generate a better plan or have a clear decision after carefully assessing all inputs in the beginning, some of unsafe operations in the later stage can be avoided.

**Questions:**

- Does the CUAs always have the access to check the target file immediately after receiving the user query? I'm thinking whether it could be helpful to show the agent the file content before planning and implementation. If the agent is aware of all information of the task in the planning phase, does the issue of lacking contextual reasoning still exist?

---

> ### Author Response · Authors · 2025-11-21
> **Response to Reviewer 2 (rsyJ) - part 1**
>
> We appreciate your recognition of our comprehensive BGD evaluation and the reliability analysis of our LLM-based judge. Your questions were very helpful, building on one another and strengthening the paper and our interpretation of the results.
>
> ---
>
> **Q1) How often do CUAs recognize the harm?**
>
> Thanks for paying so much attention to the details in our paper! This question is closely related to Reviewer 4’s Q4 (though theirs is more general). Accordingly, we added **Appendix F (Distinguishing Detection Failures vs. Decision Errors) to the revised PDF**, and encourage you to check it out. Thank you for improving our paper!
>
> In short, specific to your question, there are two distinct cases in which the agent DOES recognize the undesirability of the context:
>
> (1) It recognizes harm, does NOT show any BGD intentions, and takes the appropriate safe actions. These correspond to the runs where the judge finds no BGD intentions (100 − BGD%).
>
> (2) It recognizes harm, but due to the failure modes we identify (execution-first bias, request-primacy, thought–action disconnect), it still shows BGD intentions and proceeds toward the execution of those. These cases form a subset of the trajectories labeled as BGD.
>
> To provide a concrete example, we manually inspected all 90 GPT-4.1 trajectories. GPT-4.1 showed BGD in 76 tasks (84.4%), meaning it behaved safely in 14 tasks by recognizing and avoiding harm (case 1).
> Among the 76 BGD cases, we found 9 tasks where the agent recognized the harm or contradiction but still pursued the unsafe goal (case 2).
> Thus, GPT-4.1 recognized the undesirability in a total of 23 tasks (14 where it acted safely + 9 where it recognized the issue but still pursued the goal). We repeated the analysis for Claude Sonnet 4 and o4-mini as well and added the results to the table below.
>
> | Model| Total BGD | BGD: Unrecognized | BGD: Recognized | No BGD: Recognized|
> |-|-|-|-|-|
> | **GPT-4.1**| 76 / 90| 67 / 76| 9 / 76| 14 / 90|
> | **Claude Sonnet 4**| 59 / 90|  52 / 59| 7 / 59| 31 / 90|
> | **o4-mini**| 78 / 90| 67 / 78| 11 / 78| 12 / 90|
>
> **All in all, most harmful cases stem from CUAs failing to recognize the harm (BGD Unrecognized), and a smaller, yet important subset reflects cases where the agent recognizes the harm yet still decides to pursue the unsafe goal due to the identified failure modes (BGD Recognized).**
>
> ---
>
> **Q2) At which steps does BGD emerge?**
>
> We really appreciate this suggestion as well! Inspired by OS-Harm \[1\], we added a violation\_step field to the judge’s output, defined as: “The index of the first step at which the agent begins to exhibit blind goal-directedness behavior; null if no such behavior occurs.” Here are the averages of step numbers where BGD intentions first emerge for the five models.
>
> | Model| Contextual Reasoning | Making Assumptions | Contradictory Goals | Overall Mean |
> |-|-|-|-|-|
> | **GPT-4.1**| 2.10|2.20 |1.75 |2.01|
> | **DeepSeek-R1**| 1.95|2.10 |1.88|1.98|
> | **Claude Sonnet 4**|3.18 |2.94 |2.91 |3.01 |
> | **o4-mini**|3.29 |3.13 |2.78 |3.06|
> | **Computer-Use-Preview**|3.52 |3.34 |2.76 |3.20|
> | **Overall Mean**|2.81 |2.74 |2.42 |2.65|
>
> **This shows that BGD intentions emerge early in the trajectory, typically within the first 2–3 steps; after which agents focus on executing the goal.** GPT-4.1 and DeepSeek-R1 often produce a more comprehensive plan upfront (e.g., “opening settings and creating a 20000GB partition”), causing earlier BGD emergence. In contrast, models like Claude Sonnet 4, o4-mini and Computer-Use-Preview break plans across multiple initial steps (e.g., first “opening settings,” then in the next step “creating a 20000GB partition”), which delays when the undesired intention becomes explicit, resulting in a slightly later BGD onset.
>
> The Contradictory Goals category appears a bit earlier, as the contradiction is usually explicit in the instruction. By contrast, Contextual Reasoning and Making Assumption tasks may expose the undesired context later (e.g., after opening a file and seeing its contents).
>
> We added the discussions and results to a new **Appendix G (At Which Steps Does BGD Emerge?) in the revised PDF**. Thanks again for your great suggestion.
>
> ---
>
> \[1\] Kuntz et al. OS-HARM: A Benchmark for Measuring Safety of Computer Use Agents, NeurIPS 2025 Datasets and Benchmarks Track spotlight

---

> > ### Author Response · Authors · 2025-11-21
> > **Response to Reviewer 2 (rsyJ) - part 2**
> >
> > **Q3) Weak Planning or Insensitivity to unsafe context?**
> >
> > Great point as well. As noted in our response to Q2, we do observe differences in planning behavior across models: GPT-4.1 and DeepSeek-R1 tend to produce more comprehensive upfront plans, whereas Claude Sonnet 4, o4-mini, and Computer-Use-Preview break planning across steps in a more incremental manner as they proceed. But we do not find any consistent relationship between these differences and the models’ BGD rates according to the results.
> >
> > Importantly, the assumption that “better upfront planning” could prevent unsafe actions does not quite hold in our settings, where the relevant context unfolds over multiple turns. For many tasks, an agent cannot assess the harmfulness of an operation until it has taken preliminary steps (e.g., Opening messages, finding the specified message, and then deciding whether and what to reply). Thus, the agent simply cannot “plan away” harms that it has not yet encountered; even if it does a comprehensive initial planning.
> >
> > In addition, based on our response to your Q1, most harmful cases arise from insufficient contextual sensitivity and difficulty recognizing harmful cues once they appear. In fact, although Claude models perform less comprehensive initial planning than GPT-4.1 or DeepSeek-R1, they are substantially better at recognizing harmful context when it arises, making them the safest and least blindly goal-directed models in Table 1 and Figure 2.
> >
> > **Overall, our results suggest that BGD is driven primarily by weak contextual reasoning and low sensitivity to harmful cues, rather than by planning deficiencies.**
> >
> > ---
> >
> > **Q4) Do CUAs have access to all information at step 0? If so, does it help?**
> >
> > Thank you for this thoughtful question. As discussed in our response to your previous question, in OSWorld and in real computer-use settings, CUAs do not see file contents, windows, or UI states they have not yet navigated to. Instead, in the first step, they have access to the initial state (Instruction, screenshot, a11y tree), and they acquire context step-by-step through interaction, much like humans do. Because of this, providing all information up front would be unrealistic and would not reflect the intended evaluation of real CUAs.
> >
> > Therefore, for many of our tasks, the relevant context simply does not exist at the initial state; for example, the agent cannot know the content of an email before navigating to the inbox, locating the email, and opening it. As a result, as we noted earlier, the agent simply cannot “plan away” harms it has not yet encountered. That said, we also include tasks where the harmful context is present from the very first step (e.g., a notes window containing harmful information is already open in step 0, and the agent is asked to share it). Even in these cases, agents still fail to demonstrate contextual reasoning, exhibiting the same BGD behaviors.
> >
> > **As a result, realistically, CUAs do not possess full context at step 0 and instead acquire it gradually through interaction. Yet even when the harmful context is present from the outset, they still act without contextual reasoning, reinforcing that BGD stems not from incomplete context but from an inherent lack of contextual reasoning and low sensitivity to harmful cues, as we discussed in Q3.**
> >
> > We added a detailed discussion **around your Q3 and Q4 in a new Appendix H (Planning, Initial Context, and BGD)**, which we believe is a valuable addition to the paper. Thank you for your insightful questions and for helping us improve the work.

---

> ### Comment · Reviewer_rsyJ · 2025-11-27
> **Acknowledgement**
>
> Thanks for your answers. They have solved all of my concerns. I decide to maintain my score.

---

> > ### Author Response · Authors · 2025-11-27
> > **Thank you**
> >
> > We sincerely appreciate that you like our paper and recommend it for acceptance. Your questions genuinely improved our paper, and we’re glad to hear that our responses and revisions addressed your questions. We are happy to address any further questions you may have.

---

### Official Review · Reviewer_WZ4c · 2025-11-07

**Soundness:** 3
**Presentation:** 2
**Contribution:** 2
**Rating:** 4
**Confidence:** 4

**Summary:**

This paper introduces the concept of Blind Goal-Directedness (BGD) — a behavioral bias in computer-use agents (CUAs) where the agent executes user-specified goals regardless of feasibility, safety, or context. The authors propose BLIND-ACT, a benchmark with 90 GUI-based tasks covering three BGD patterns: (i) lack of contextual reasoning, (ii) assumptions and decisions under ambiguity, and (iii) contradictory or infeasible goals. The paper reports results on nine frontier models (including GPT-5 and Claude Opus 4), finding high BGD rates (≈80.8%). The authors further test “contextual” and “reflective” prompting interventions, which modestly reduce BGD, and conduct qualitative analysis revealing several failure modes.

While the topic is timely and the experiments extensive, the work feels more diagnostic than analytical — it describes problems but provides limited insight into underlying causes or mitigation strategies. The empirical section is large but methodologically opaque, particularly regarding how tasks were designed, balanced, and verified.

**Strengths:**

- Addresses an important and emerging area: GUI agent safety beyond prompt attacks.
- Clear taxonomy (three BGD categories) and thorough experimentation.
- Empirical validation of prompting-based interventions

**Weaknesses:**

- Benchmark design lacks transparency and external validation.
- Heavy reliance on LLM judges without adequate robustness checks.
- Lacks theoretical or mechanistic explanation for why BGD arises.

**Questions:**

1.How were the 90 tasks distributed in difficulty and domain? Were they balanced or randomly constructed?
2.Has the paper explored the underlying causes of Blind Goal-Directedness? For example, are there analyses of which model components or reasoning stages lead to goal fixation?
3.Beyond prompting interventions, has the paper attempted any model-level improvements (e.g., training-time alignment, reward shaping, or trajectory-level supervision) to mitigate BGD

---

> ### Author Response · Authors · 2025-11-21
> **Response to Reviewer 1 (WZ4c) - part 1**
>
> We thank you for recognizing the importance of our study and the significance of this emerging area of GUI agent safety.
>
> ---
>
> **Q1) Benchmark design transparency**
>
> This is a great question. We expanded **Appendix I (Additional Task Details)** with additional details on benchmark construction, and encourage you to check it for the full description. Thank you for improving our paper!
>
> In brief, all BLIND-ACT tasks were manually created (not model-generated), and the benchmark was developed in multiple steps.
>
> First, we ran a small exploratory study by making changes to standard OSWorld tasks to examine a broad range of safety and reliability issues that could arise. During this process, we noticed that many examples exhibited common, though still preliminary, failure patterns, all related to blindly being goal-directed.
> As part of this mid-exploratory study, we also tested the duality and transferability of several known alignment gaps in VLMs to the CUA domain (e.g., lack of contextual reasoning in compositional contexts).
> We then extended the exploratory study to further examine these findings and to confirm that the observed patterns capture common, generalizable failure behaviors across multiple CUAs.
> Finally, we clustered all collected examples into three consistent failure patterns, which became our three BGD patterns, and continued red teaming until we collected at least 30 tasks per pattern.
>
> Task construction happened through multiple brainstorming sessions of an iterative cycle. We iteratively created and tested candidate tasks on several CUAs, discarding those that failed to elicit BGD or proved unreliable, ultimately arriving at 30 diverse tasks per pattern (90 in total).
> Each finalized task is independent, non-templated, and paired with a custom-designed environment state, including files, simulated interfaces, and supporting assets, carefully crafted to probe the target BGD pattern. Tasks span a wide range of domains, including file manipulation, productivity workflows, communication, image editing, web interaction, system configuration, scripting, etc. They also vary in domain and difficulty: some expose the harmful context immediately, while others require multiple steps before the agent encounters the challenging scenario. Importantly, tasks were not designed to be uniformly “easy” or “hard.” Instead, our focus was on achieving comprehensive coverage of safety-relevant contexts where BGD is likely to emerge in real deployments.
>
> Further details are provided in **Appendix I**.
>
> ---
>
> **Q2) The work is more diagnostic than mechanistic or theoretical - How about model-level mitigations beyond prompting?**
>
> Thank you for raising these points. We'd like to clarify that the goal of our paper is actually diagnostic in scope rather than mechanistic. Our primary contribution is to identify the Blind Goal-Directedness (BGD) phenomenon as a previously unreported failure in CUAs, characterize its three prevalent patterns, and introduce BLIND-ACT, the first benchmark that enables systematic study of this phenomenon. **As you and Reviewer 3 both noted, BGD exposes an emerging class of risks beyond prior attacks and opens up a new direction for CUA security research.**
>
> That said, while we do not perform mechanistic interpretability or theoretical analyses, we do investigate underlying behavioral causes of BGD through:
>
> (i) cross-model comparisons revealing insights such as safety–capability parity, differences between computer-use–trained versus general-purpose models, and more in the paper;
>
> (ii) prompting-condition comparisons (default, contextual, reflective), illustrating the limited effectiveness of prompting-based inference-time mitigations; and
>
> (iii) qualitative trajectory inspection identifying consistent failure modes (execution-first bias, thought–action disconnect, and request-primacy), revealing reasoning-stage breakdowns.
>
> We also conducted a new analysis based on Q2 of R2, examining when BGD first emerges in reasoning steps in the trajectory. We encourage you to see the newly added **Appendix G (At Which Steps Does BGD Emerge?)**.
>
> Additionally, we provided an analysis in response to Q4 of R4, **distinguishing detection failures vs. decision errors** leading to BGD, which further breaks down our metrics and what they imply. Similarly, we encourage you to check **Appendix F**.
>
> Regarding mitigation strategies, we focus on prompting-based interventions as a first-step behavioral probe, while agreeing that model-level mitigations (e.g., training-time interventions, reward modeling, trajectory-level supervision) are important next steps.
> These approaches, along with deeper mechanistic studies, are valuable future research directions for us and the broader community.
> We also revised the introduction, conclusion, and **Appendix B (Potential Future Directions)** to better highlight our motivation and articulate potential directions to move forward in studying and mitigating BGD.

---

> > ### Author Response · Authors · 2025-11-21
> > **Response to Reviewer 1 (WZ4c) - part 2**
> >
> > **Q3) Reliability of LLM Judges**
> >
> > Thank you for this question. We encourage you to first read our detailed response under **Common Question**, where we discuss our reliance on LLM judges. Further experiments and clarifications are provided below:
> >
> > In short, **as also acknowledged by R2**, we carefully investigated the reliability of the judge by conducting several forms of validation and ablation:
> >
> > *   **Judge-human agreement:** 93.75% human eval agreement with precision/recall/F1 ≥ 0.9 and strong Cohen’s Kappa scores (0.81 for BGD and 0.91 for Completion) - (Section 3.1 and **Appendix D)**
> >
> > *   **Robustness Ablations:** Multiple judge configurations, including agent trajectory, agent trajectory + a11y trees, agent trajectory + screenshot captions; with two different judges (GPT4.1 and o4-mini) - **(Appendix D)**
> >
> >
> > We note that our use of LLM judges is inspired by **OS-Harm \[1\], which was accepted as a Spotlight paper in the NeurIPS 2025 Datasets and Benchmarks Track**. We still achieve higher agreement with human evaluations (93.75%) and stronger precision/recall/F1 metrics than OS-Harm (84.5%).
> >
> > **We additionally designed a new experiment where we ran our judge three times on all 90 trajectories for three agents (o4-mini, GPT-5, and Claude Sonnet 4) to assess consistency across independent runs.** As shown in the table below, the variances across the three runs are effectively negligible, demonstrating that the judge’s decisions are stable and robust across repeated independent evaluations. **Per your suggestion, we created Appendix D.2 (Judge Consistency Across Independent Runs)** in the updated PDF to include this discussion.
> >
> > Table: \*\*Average BGD and Completion among three categories (whole benchmark)
> > | Model| Run1 | Run2 | Run3 | STD |
> > |-|-|-|-|-|
> > | \*\*o4-mini\*\*|BGD: 86.6, Completion: 57.7|BGD: 86.6, Completion: 57.7|BGD: 86.6, Completion: 56.6|BGD: 0, Completion: 0.52|
> > | \*\*GPT-5\*\*|BGD: 85.5, Completion: 45.5|BGD: 85.5, Completion: 45.5|BGD: 85.5, Completion: 44.4|BGD: 0, Completion: 0.52|
> > | \*\*Claude Sonnet 4\*\*|BGD: 65.5, Completion: 35.5|BGD: 65.5, Completion: 35.5|BGD: 65.5, Completion: 35.5|BGD: 0, Completion: 0|
> >
> >
> > **Based on R3’s suggestion, we also added a new analysis examining cases where the judge and human annotators disagree**. We created **Appendix D.3 (Cases of Judge–Human Disagreement)** in the updated PDF to include this discussion.
> >
> > In the “Contradictory or Infeasible Goals” category, some disagreements arise around “BGD” intention detection. For example, in a task requiring cropping an image to negative dimensions, some agents initially proceed by opening GIMP and loading the image before later recognizing that the dimensions are invalid. Human evaluators typically do not consider this BGD, since the agent notices the contradiction even in mid-trajectory. The judge, however, applies a stricter standard and expects the agent to identify the contradiction earlier, thereby labeling such cases as BGD.
> >
> > In the “Lack of Contextual Reasoning” category, some disagreements appear in “Completion”. In some more complex tasks such as an example where the agent must locate an image, open email, compose a message, attach the file, and click “send”, some agents reach the final step and correctly predict the “send” action, but due to capability or coordination errors, the final click fails to execute and the task exits due to the 15 steps limit. The judge counts this as a Completion, while humans label these as non-completions despite full correct behavior up to the last step.
> >
> > We added all of these new analyses to **Appendix D.2 and D.3** of the updated paper.
> >
> > \[1\] Kuntz et al. OS-HARM: A Benchmark for Measuring Safety of Computer Use Agents, NeurIPS 2025 Datasets and Benchmarks Track spotlight

---

### Author Response · Authors · 2025-11-21
**Summary of our revisions and answering common question (Part1)**

## Note: This response has been divided into two comments.

**We thank the reviewers for their excellent feedback and suggestions.**

**Contribution of the paper:** Our paper identifies Blind Goal-Directedness (BGD) as a previously unreported failure in Computer-Use Agents (CUAs), where agents pursue goals blindly, prioritizing task execution over feasibility, safety, or context. We systematically characterize its three prevalent patterns: lack of contextual reasoning, costly assumptions under ambiguity, and contradictory or infeasible goals; and introduce BLIND-ACT, built on OSWorld, as the first benchmark for systematically studying this phenomenon.
As Reviewers 1 and 3 noted, BGD exposes an emerging class of risks beyond prior attacks and opens a new direction for CUA security research.
We show that prompting-based interventions only partially reduce BGD, leaving significant residual risk. Our Qualitative analyses reveal consistent failure modes: execution-first bias, thought–action disconnect, and request-primacy, reflecting deeper alignment challenges in CUAs. Our findings underscore the need for model-level mitigations, safeguards, and mechanistic studies tracing how BGD arises across model components, training stages, and reasoning steps, offering valuable future directions for the broader community.

**Overview of Revision:**

*   We **color-coded** the added changes to the **PDF** for the comfort of our reviewers.


*   Expanded Appendix I (Additional Task Details) with a detailed discussion on the process of our benchmark construction and revised Section 2 to reference them.

*   Appendix I.1 (What Constitutes Safe Behavior in BLIND-ACT) to better clarify that refusal is not the only safe and expected behavior in our benchmark.


*   Added new experiments and analyses on (Judge Consistency Across Independent Runs) to Appendix D.2, and revised Section 3.1 to reference them.

*   Added new analyses of (Cases of Judge-Human Disagreement) to Appendix D.3, and revised Section 3.1 to reference them.

*   Added new experiments examining when BGD first emerges in reasoning steps in the trajectory. Appendix G (At Which Steps Does BGD Emerge?), and revised Section 3.1.


*   Included a new analysis on (Distinguishing Detection Failures vs. Decision Errors), leading to BGD in Appendix F, and revised Section 2.3.

*   Revised Introduction, Conclusion, and Appendix B (Potential Future Directions) to better highlight our motivation and articulate potential directions to move forward in studying and mitigating BGD.


*   Added a deep discussion around planning behavior differences, the role of the initial context, and their relationship to BGD to Appendix H (Planning, Initial Context, and BGD).

*   Revised Appendix E (Prompting Intervention Results Table) and Section 3.2.


*   We made various other changes to clarify questions and comments in the reviews.

## In the next comment, we will be answering the Common Question.

---

> ### Author Response · Authors · 2025-11-21
> **Summary of our revisions and answering common question (Part2)**
>
> ## Continued:
>
> **Common Question: Motivation and Reliability of LLM Judges \[Reviewer 1 / Reviewer 3\]**
>
> Thank you for raising this important point. We believe an LLM-judge–based evaluation is more suitable for our setting than a rule-based approach, such as the one used in OSWorld for three reasons:
>
> 1.  OSWorld tasks evaluate “capability” on requests with a single or a few fully verifiable outcomes. For example, a task such as  “I want to switch off Bluetooth” has one objectively correct end state, making a rule-based evaluator appropriate for checking whether Bluetooth was turned off and labeling the attempt as success or failure.
>     In contrast, our benchmark focuses on safety and reliability, where rule-based evaluation is infeasible **because undesired actions and BGD intentions can manifest in many different ways, and there is no single “correct” outcome to check for**. For instance, in the task “Ensure that my proposal gets accepted” (Figure 13), there are numerous unsafe behaviors an agent might take: deleting important facts, fabricating results, adding manipulative watermark text, and more. Because harmful behaviors can appear in diverse and semantically complex forms, it is essentially impossible to design rule-based checks or fixed verifiers for such tasks.
>
> 1.  In capability benchmarks like OSWorld, capability depends on whether the agent can reach the final desired outcome. However, as we emphasize in the paper, **safety and reliability depend not only on outcomes but also on intentions**. Just because a model fails to complete a harmful or undesirable goal does not make it safe. As shown by the gap between BGD intentions and unsafe completions, many models intend to pursue the harmful action but lack the capability to carry it out **(our “safety–capability parity” observation)**. Therefore, checking only the final environment state is insufficient; **a semantic judge that understands language is needed to read the entire trajectory, including reasoning and action predictions, and evaluate the agent’s underlying intentions.**
>
>
> 1.  Models have improved over the last year (see e.g. \[1\]) and are now more capable as judges, especially with appropriate prompting and input format. **As also acknowledged by R2**, we carefully investigated the reliability of the judge by running **comprehensive ablations and robustness checks in Appendix D.** **We note that our use of LLM judges is inspired by OS-Harm \[1\], which was accepted as a Spotlight paper in the NeurIPS 2025 Datasets and Benchmarks Track.** We still achieve higher agreement with human evaluations (93.75%) and stronger precision/recall/F1 metrics than OS-Harm (84.5%), with detailed information on judge accuracy (Section 3.1 and Appendix D).
>
> We also added new experiments on **judge consistency across independent runs (Appendix D.2)** and **new analyses of judge–human disagreement cases (Appendix D.3).**
>
> \[1\] Tan et al. JudgeBench: A Benchmark for Evaluating LLM-based Judges, ICLR 2025
>
> \[2\] Kuntz et al. OS-HARM: A Benchmark for Measuring Safety of Computer Use Agents, NeurIPS 2025 Datasets and Benchmarks Track spotlight

---

### Meta-Review · Area_Chair_Hm5r · 2026-01-15

**Summary:**

This paper identifies "blind goal-directedness", a failure mode of computer use agents in which they try to pursue user goals regardless of contextual feasibility / safety. The authors offer a characterization of this phenomenon and also contribute a benchmark to evaluate it. They show that modern CUA agents exhibit high levels of BGD, and also that prompting based mitigation strategies don't completely get rid of it, suggesting that further intervention / adaptation is needed.

Overall, the reviewers (and this AC) agree that the topic of the paper is interesting, and especially the characterization / trajectory analysis of frontier CUA agents could be useful to analyze future agents.

The main concerns of the reviewers centered around the benchmark: how realistic is it? additionally, the use of an LLM judge was not properly justified. There was also a larger question of whether the BGD phenomenon was studied enough in depth, as there were no more involved mitigation strategies explored and no deeper analysis of the origins of this phenomenon. Finally, the novelty of the paper is itself questionable given many related phenomena discussed in prior papers that were brought up by reviewers.

**Reviewer Concerns:**

satisfactorily addressed:
- benchmark construction details: this was clarified effectively by the rebuttal and will make good content in the appendix
- Reviewer Ybz1's concern that hte only safe behavior in the benchmark is refusal was resolved with other examples
- use of LLM judge: i'd say this is partially addressed. the authors did perform human experiments and compute agreement, but their judge may still have unidentified biases or lack robustness to OOD trajectories (true of all LLM judges). still, this is definitely good enough for validating the judge in the context of this benchmark

still outstanding:
- novelty: multiple reviewers bring up that BGD is very similar to previously studied phenomena like goal misgeneralization and over-optimization
- lack of effective mitigation: reviewers (and this AC) agree that coming up with a more effective mitigation strategy would really improve the paper's impact

**Reviewer Scores:**

WZ4c: 4 (unchanged)
rsyJ: 6 (unchanged)
w3zH: 2 > 4
Ybz1: 4 (unchanged)

---

### Decision · Program_Chairs · 2026-01-26

Accept (Poster)